# TRAP1 in Oxidative Stress and Neurodegeneration

**DOI:** 10.3390/antiox10111829

**Published:** 2021-11-19

**Authors:** Inês Ramos Rego, Beatriz Santos Cruz, António Francisco Ambrósio, Celso Henrique Alves

**Affiliations:** 1Retinal Dysfunction and Neuroinflammation Lab, Faculty of Medicine, Coimbra Institute for Clinical and Biomedical Research (iCBR), University Coimbra, 3000-548 Coimbra, Portugal; uc2018273660@student.uc.pt (I.R.R.); 20180221@esac.pt (B.S.C.); afambrosio@fmed.uc.pt (A.F.A.); 2Centre for Innovative Biomedicine and Biotechnology (CIBB), University Coimbra, 3004-531 Coimbra, Portugal; 3Faculty of Sciences and Technology, University Coimbra, 3030-790 Coimbra, Portugal; 4Association for Innovation and Biomedical Research on Light and Image (AIBILI), 3000-548 Coimbra, Portugal; 5Clinical Academic Center of Coimbra (CACC), 3004-561 Coimbra, Portugal

**Keywords:** HSP75, HSP90, mitochondria, neurodegeneration, oxidative stress, TRAP1

## Abstract

Tumor necrosis factor receptor-associated protein 1 (TRAP1), also known as heat shock protein 75 (HSP75), is a member of the heat shock protein 90 (HSP90) chaperone family that resides mainly in the mitochondria. As a mitochondrial molecular chaperone, TRAP1 supports protein folding and contributes to the maintenance of mitochondrial integrity even under cellular stress. TRAP1 is a cellular regulator of mitochondrial bioenergetics, redox homeostasis, oxidative stress-induced cell death, apoptosis, and unfolded protein response (UPR) in the endoplasmic reticulum (ER). TRAP1 has attracted increasing interest as a therapeutical target, with a special focus on the design of TRAP1 specific inhibitors. Although TRAP1 was extensively studied in the oncology field, its role in central nervous system cells, under physiological and pathological conditions, remains largely unknown. In this review, we will start by summarizing the biology of TRAP1, including its structure and related pathways. Thereafter, we will continue by debating the role of TRAP1 in the maintenance of redox homeostasis and protection against oxidative stress and apoptosis. The role of TRAP1 in neurodegenerative disorders will also be discussed. Finally, we will review the potential of TRAP1 inhibitors as neuroprotective drugs.

## 1. Introduction 

Tumor necrosis factor receptor-associated protein 1 (TRAP1), also known as heat shock protein 75 (HSP75), is a member of the heat shock protein 90 (HSP90) chaperone family that resides mainly in the mitochondria. In humans, the HSP90 family is composed of three more isoforms, namely HSP90α and HSP90β located in the cytosol and glucose-regulated protein 94 (GRP94) present in the endoplasmic reticulum [1]. The prototypic HSP90 family member has three structural domains, namely, an N-terminal domain (NTD) responsible for ATP binding, a middle domain (MD) which is important for binding clients and the ɣ-phosphate of ATP, and a C-terminal domain (CTD) responsible for dimerization. Like other chaperones, HSP90 family members are important for the folding of client protein substrates, as well as the degradation of the misfolded proteins [2,3]. 

TRAP1 was extensively studied in oncology once TRAP1 is selectively upregulated in several human tumors and this is correlated with malignant progression, metastatic potential, and resistance to chemotherapy [3]. For those reasons, TRAP1 is emerging as a therapeutic target with several researchers attempting to develop TRAP1 specific inhibitors. Mitochondrial TRAP1 is a cellular regulator of oxidative stress-induced cell death [4,5], mitochondrial homeostasis and bioenergetics [6] and unfolded protein response (UPR) in the endoplasmic reticulum (ER) [7,8]. Although, all the knowledge about the role of TRAP1 in tumor cells, its role in central nervous system cells, under physiological and pathological conditions, remains largely unknown.

Here, we will describe the structure and the functions of TRAP1 in the cell, with particular attention to its role in the maintenance of redox homeostasis. Furthermore, we will also explore the role of TRAP1 in neurodegenerative disorders and the potential of TRAP1 inhibitors as neuroprotective drugs. 

## 2. TRAP1 Molecular Structure 

The human *TRAP1* gene locus resides on chromosome 16p13, spans 60 kb, and includes 18 exons. The main *TRAP1* transcript composed of 2,112 bp, encodes a protein of 704 amino acids [9] containing three major domains: NTD, MD and CTD (Figure 1), as previously mentioned [2,3,10]. 

TRAP1 has an N-terminus 59 amino acid sequence serving as mitochondrial targeting that is cleaved off upon the protein being translocated to the mitochondria [10,16]. TRAP1 is expressed in several tissues, including the central nervous system (https://www.proteinatlas.org/ENSG00000126602-TRAP1/tissue; accessed on 7 November 2021) [17,18,19].

Unlike other HSP90 paralogs, both TRAP1 and its bacterial homologue chaperone protein HtpG (HtpG) [20] lack the charged linker between the MD and CTD [21] and do not have any cochaperones [22,23]. Another distinctive aspect is that TRAP1 features a long extension of the N-terminal β-strand that crosses between monomers in the closed state, acting as a thermal regulator of protein function [9]. TRAP1 also shows a marked asymmetric conformation, due to reconfigurations in the MD:CTD interface, critical for client binding [24]. Interestingly, TRAP1 can form tetramers, as dimer of dimers [25]. Although ATP turnover rate activity is comparable to other HSP90 paralogs [26], the affinity of TRAP1 for ATP is one order of magnitude higher [27]. Indeed, the Michaelis constant (K_m_) of TRAP1 for ATP binding is 14.3 µM, much lower than the average K_m_ of 127 µM for the other HSP90 paralogs [27]. Having an ATPase activity, the chaperone TRAP1 regulates substrate influx and takeover through the ATPase cycle, following a distinct mechanism than other HSP90 chaperones [28]. Research on the ATPase cycle of TRAP1 showed that both TRAP1 protomers undergo structural changes through rounds of ATP binding, hydrolysis, and release [27,29,30,31,32]. These conformational changes induced by ATP may influence the conformation of the bound client protein [33]. On its ATPase cycle, TRAP1 can adopt three different conformations: an open state, an intermediate coiled-coil state with proximate N-terminal domains, and a closed state in which the N-terminal extends between protomers [31]. Upon ATP binding, without the additional action of co-chaperones, TRAP1 undergoes a dramatic structural change, adopting a closed asymmetric conformation [27,29,34]. The energy required for client remodeling is given by the two-step ATP hydrolysis [29]. The first step of ATP hydrolysis is responsible for the rearrangement of the binding site of client molecules [31,32]. Following this step, the second ATP hydrolysis causes conformational changes in TRAP1, that results in the release of client molecules [31]. In summary, the first hydrolysis reaction couples client remodeling, whereas the second would lead to its dissociation and the return of TRAP1 to the apo state [24]. The TRAP1 ATPase pocket can also bind calcium (Ca^2+^) instead of magnesium (Mg^2+^) as an enzymatic cofactor [23]. While Mg^2+^ binding causes sequential noncooperative hydrolysis [23], when bound to Ca^2+^, TRAP1 switches its hydrolytic activity to a cooperative mode. Under these conditions, TRAP1 directly moves from closure to full opening, thus skipping the asymmetry flipping and client remodeling that characterizes its normal chaperone cycle [24]. The relative preference of TRAP1 for Ca^2+^ or Mg^2+^ is dependent upon the free ATP concentration, with higher ATPase activity with Ca^2+^ at high ATP concentrations (350 µM) and higher ATPase activity with Mg^2+^ at low ATP concentrations (35 µM) [23] (Figure 1).

## 3. TRAP1 Functions and Signaling Pathways

### 3.1. Role of TRAP1 in Mitochondria

#### 3.1.1. TRAP1 and Energetic Metabolism Regulation

The mitochondrial chaperone TRAP1 is an important modulator of the mitochondrial energy metabolism including oxidative phosphorylation (OXPHOS) processes [9,31,35,36], and it operates by interacting with proteins of the mitochondrial electron transport chain (ETC), as well as succinate dehydrogenase (SDH) and cytochrome oxidase, also known, respectively, as complex II and complex IV. It also acts as a key regulator of the protein quality control (PQC) [2,9,25,31]. 

The interaction between TRAP1 and the mitochondrial proto-oncogene tyrosine-protein kinase (c-Src) downregulates cytochrome oxidase activity [35,37,38]. TRAP1 also regulates the ability of mitochondrial c-Src to stimulate OXPHOS [37,38]. TRAP1 interaction with subunit A of SDH inhibits SDH activity, leading to a decrease in reactive oxygen species (ROS) generation [2,6]. The inhibition of SDH activity is enhanced by TRAP1 ERK1/2-dependent phosphorylation and has antioxidant and antiapoptotic effects in tumor cells, stabilizing the hypoxia-inducible factor 1α (HIF1α), a transcription factor required for tumor cell growth [31,36,39]. TRAP1-dependent stabilization of HIF1α [36] occurs independently of oxygen availability and induces a complex transcriptional program that further shapes cell metabolism by supporting a metabolic shift toward aerobic glycolysis and repressing OXPHOS [39,40,41,42]. Supporting the inhibitory relation between TRAP1 and SDH, Sciacovelli et al. reported that TRAP1 expression was inversely correlated to SDH activity in human osteosarcoma SAOS-2 cells [36] (Figure 2). 

Several authors report that, upon the Warburg effect, TRAP1 induces the upregulation of glycolysis over OXPHOS, decreasing ROS production [9,36,43]. OXPHOS sustains organelle function and plays a central role in cellular energy metabolism, with increased energy efficiency when compared to glycolysis. Therefore, OXPHOS downregulation results in the decrease of mitochondrial respiration. Moreover, there is more NADPH available, which is an important ROS scavenger, thereby reducing ROS even further [2]. As a result, OXPHOS downregulation is associated with increased mitochondrial tolerance to oxidative stress and protection from apoptosis [31,44]. Human colorectal carcinoma HCT116 cells express higher levels of TRAP1 in comparison with surrounding healthy cells, showing a predominant Warburg phenotype [36], which might represent an advantage concerning cellular survival and proliferation. 

To clarify the function of TRAP1 in the energetic metabolism regulation, a *Trap1* knockout (*Trap1*^−/−^) mouse was generated [45]. Interestingly, *Trap1*^−/−^ mouse embryonic fibroblasts (MEFs) exhibited impaired mitochondrial function with significantly increased production of ROS and increased sensitivity to oxidative damage compared to that of wild-type (WT) cells. The homozygous deletion of *Trap1* also resulted in decreased SDH expression, reflecting loss of protein-folding quality control in mitochondria [45]. *Trap1*^−/−^ mice showed a reduced incidence of age-associated pathologies, including inflammatory tissue degeneration and spontaneous tumor formation. The animals with homozygous deletion of TRAP1 also presented global reprogramming of cellular bioenergetics, with compensatory upregulation of both oxidative phosphorylation and glycolysis transcriptomes, impairing mitochondrial respiration and cell proliferation and increasing susceptibility to oxidative stress. This was also accompanied by increased mitochondrial accumulation of cytoprotective chaperones HSP90 and HSP27 and a switch to glycolytic metabolism in vivo [45].

#### 3.1.2. TRAP1 and Redox Homeostasis: Protection against Oxidative Stress and Apoptosis

Mitochondrial dysfunction and redox homeostasis disruption mainly result from the oxidative stress induced by the accumulation of ROS produced in a side reaction of the ETC [46,47,48]. TRAP1 has a protective role against mitochondrial dysfunction, decreasing the production and accumulation of ROS, and therefore reducing oxidative stress [31,49,50,51,52]. While *TRAP1* overexpression leads to a decrease in ROS accumulation [53,54,55], *TRAP1* silencing leads to increased susceptibility to oxidative stress, showing an inverse correlation between total cellular ROS levels and TRAP1 expression [28,35,53,54,56]. TRAP1 antioxidant activity is intimately linked to its ability to control respiratory complexes of the ETC, namely, SDH and cytochrome oxidase [35,36,39]. 

Accumulation of ROS can result in the release of cytochrome C (CytC), inducing caspase-mediated apoptosis, a process of programmed cell death crucial for cell and tissue homeostasis [57]. The link between TRAP1 and apoptosis was first reported by Hua et al., who demonstrated the protective role of TRAP1 against granzyme M (GzmM)-mediated apoptosis [53]. The protease GzmM induces cell death by client proteins cleavage and initiates ROS generation and CytC release, causing mitochondrial swelling and loss of mitochondrial transmembrane potential [53]. This release induces caspase-mediated apoptosis, activating the mitochondrial apoptotic machinery. The same study also reported that silencing *TRAP1* through RNA interference increases ROS accumulation and enhances GzmM-mediated apoptosis, whereas *TRAP1* overexpression attenuates both ROS production and GzmM-mediated apoptosis [53].

Mitochondrial matrix Ca^2+^ content is known to be a key factor for the regulation of the transitions between an open and closed state of the mitochondrial permeability transition pore (mPTP) [58,59,60,61]. Mitochondrial matrix Ca^2+^ overload induces mPTP opening, even though additional factors that are only partially characterized can contribute to pore induction [59]. Since mPTP opening results in mitochondrial inner membrane potential loss, membrane rupture and apoptosis [2,10,57], mPTP regulation is critical for mitochondrial homeostasis [56]. The mitochondrial matrix protein cyclophilin D (CypD) is known to be a regulator of the mPTP [44,49,59]. Indeed, CypD-deficient mitochondria generate fewer ROS and are less susceptible to Ca^2+^-induced mitochondrial swelling and permeability transition. Additionally, the absence of CypD protects neurons from oxidative stress-induced cell death [39,44] (Figure 3).

Activated CypD sensitizes mPTP to opening, leading to mitochondrial swelling and depolarization, with outer membrane ruptures causing the release of CytC into the cytosol, and eventually, cell death [10,62,63]. PTEN-induced kinase 1 (PINK1) depletion leads to CytC release, which correlates with the reduction in TRAP1 phosphorylation by PINK1 [64]. Through its interaction with CypD, TRAP1 inhibits mPTP formation [63]. Overexpression of TRAP1 blocks the mitochondria-mediated apoptotic cascade and the ensuing caspase-3 activation [5,62]. The inhibition of the mPTP opening can be achieved indirectly through the modulation of ROS concentration [44,49].

Xiang et al., using a hypoxic model of cardiomyocytes, reported that TRAP1 silencing induced mPTP opening [4], CytC release from mitochondria into the cytosol and elevated caspase-3 activation leading to apoptosis [65], while TRAP1 upregulation had the opposite effect [4,65]. To assess TRAP1 protective role by mPTP opening inhibition, Guzzo et al. performed a whole-cell Ca^2+^ retention capacity assay, which allows a quantitative assessment of mPTP induction, and reported that *Trap1* knockdown increased mPTP sensitivity to Ca^2+^ in cancer cells, while *Trap1* overexpression inhibited mPTP opening in mouse embryo fibroblasts (MEF) cells and protected SAOS-2 and MEF cells from mPTP opening in starvation conditions [6]. Interestingly, TRAP1 inhibition of mPTP opening was observed not only in tumor cells [6,26] and hypoxic cellular models [4,65] but also in NRK-52E kidney cells under high glucose conditions prompting oxidative stress [66], in H9C2 myocardial cells exposed to extracellular acidification [67] and in C17.2 neural stem cells [5]. 

Guzzo et al. also reported that *Trap1* knockdown by short hairpin TRAP1 RNA (shTRAP1) in SAOS-2 and human cervix carcinoma HeLa cells caused a constitutive increase in the levels of intracellular ROS and of mitochondrial superoxide anion [6]. When TRAP1 expression was rescued in shTRAP1 cells, both global ROS and mitochondrial superoxide levels returned to basal values. Cytofluorimetric analysis of mitochondrial superoxide levels with the MitoSOX probe in SAOS-2 and MEF cells showed that TRAP1 expression decreases intracellular ROS levels, protecting cells from oxidative stress and death under conditions of serum and glucose depletion [6]. Notably, *Trap1* overexpression in nontransformed MEFs downregulated intracellular ROS and mitochondrial superoxide anion levels [6].

This highlights the pathophysiological relevance of TRAP1 in the maintenance of mitochondrial and redox homeostasis under specific stress conditions [24], such as oxidative stress damage, and protection against mitophagy and apoptosis.

### 3.2. TRAP1 Suggested Extra-Mitochondrial Roles

Several research groups reported TRAP1 extramitochondrial locations and interactions, namely on the endoplasmic reticulum (ER) membrane and in the cytosol [7,21,49,62,68]. However, there is a wide controversy on the possibility that TRAP1 could act outside mitochondria. Although the mechanisms are not fully elucidated yet, interactome data also suggest that a considerable number of the client proteins of TRAP1 do not reside in the mitochondria [2,10]. As further data are required to surely determine TRAP1 extra-mitochondrial roles, here we discuss TRAP1 possible functions outside the mitochondria accordingly to the studies already performed in this field.

Besides being responsible for protein synthesis, folding and transport, ER also functions as a critical apoptosis control point. The ER homeostasis might be affected by many stress conditions, namely, changes in the oxidative environment that induce ER stress [60]. Under ER stress conditions, the ER function might be restored by the unfolded protein response (UPR) signaling pathway. However, under conditions of continuous excessive ER stress, the UPR might also trigger apoptosis [69]. Since the ER-mitochondria interface exchanges molecules for apoptosis induction, the interaction between the ER and the mitochondria is crucial for cell death regulation [70]. The most prominent signal between these organelles is Ca^2+^, which mediates the induction of apoptosis at high concentrations [70]. Interestingly, TRAP1 is implied to be involved in Ca^2+^ communication [2,7,49,71]. Moreover, the antiapoptotic activity of ER-associated TRAP1 is closely related to the Ca^2+^-binding protein Sorcin [72]. By controlling the quality of specific mitochondrial client proteins, TRAP1 is involved in the regulation of the mitochondrial apoptotic pathway and might also have a role in ER stress prevention [73].

Amoroso et al. described the presence of TRAP1 at the interface between ER and mitochondria [7]. Nevertheless, it is important to highlight that precise analyses are essential to separate ER and mitochondria components in subcellular fractionation experiments, thus discriminating TRAP1 exact localization outside the mitochondria. The authors also suggested that the proteasome regulatory tat-binding protein 7 (TBP7) is a putative TRAP1-interacting protein involved in the control of cellular ubiquitination and the quality control of misfolded mitochondrial proteins, namely, Sorcin isoform B and F1ATPase β subunit [7]. They also showed, by biochemical and confocal/electron microscopic analyses, that TRAP1 and TBP7 colocalize and interact directly in the ER. TRAP1 silencing in HCT116 human colon cancer cells exposed to ER stress enhanced stress-induced cell death and increased intracellular protein ubiquitination, thus suggesting a role of TRAP1 in ER stress protection and the quality control of specific mitochondrial proteins contributing to the regulation of the mitochondrial apoptotic pathway. This phenotype of increased total ubiquitination levels is rescued by the transfection of an extramitochondrial TRAP1 mutant [7,49]. The proposed interaction network between ER-localized TRAP1 and TBP7 might provide a novel model of the ER-mitochondria crosstalk [7] (Figure 4).

Takemoto et al. suggested a correlation between mitochondrial TRAP1 and the UPR in the ER. Their findings show that, under conditions of *TRAP1* knockdown, there was an increase in the expression of ER-resident caspase-4, which is activated by ER stress [8]. Under these conditions, they also reported an increase in the expression of 78 kDa glucose-regulated protein/binding immunoglobulin protein (GRP78/BiP), a stress-responsive ER chaperone, and a central regulator of the UPR that protects cells against ER stress-induced apoptosis [60]. There was also a decrease in the expression of C/EBP homologous protein (CHOP), which induces cell death, even under ER stress [8,55,60]. However, in comparison to that of control cells, *TRAP1* knockdown cells did not show a significant increase in the level of cell death after the early phase of ER stress [8]. Although these correlations suggest that mitochondrial TRAP1 might be a potential regulator of the UPR in the ER [8], further experiments may be needed to surely determine the exact role of TRAP1 in the ER. 

Further research reported that TRAP1 might have a role in translational processes, modulating phosphorylation levels, both in basal conditions and under ER stress, of the translation factor eIF2α [28]. Under conditions of *TRAP1* knockdown, HCT116 cells present a decrease in eIF2α phosphorylation, the initial response to ER stress, blocking cap-dependent translation. If continuous, this blockade of protein synthesis might induce cell death. Under *TRAP1* knockdown, there is also a decrease in the activation of selective stress-responsive protein RNA-like ER kinase (PERK), a key component in UPR signaling [28]. Accordingly, when *TRAP1* is silenced, HCT116 cells are more susceptible to oxidative damage, enhancing ER stress. Nevertheless, these extra-mitochondrial effects of TRAP1 in eIF2α phosphorylation and ER stress proteins are merely correlative and thus, further data may be necessary to determine the mechanistic connections in which TRAP1 is involved.

Studies in human breast adenocarcinoma MCF7 cells showed that ER stress-resistant cell lines are characterized by the upregulation of TRAP1 and GRP78 [49]. Moreover, the disruption of the ER-associated TRAP1/TBP7 pathway reestablished drug sensitivity in drug-resistant cells. The downregulation of TRAP1 in combination with ER stress agents produced high cytotoxic effects in MCF7 cells, unable them to induce the phosphorylation of PERK in response to damaging agents. These results suggest that ER-associated TRAP1 may play a role against DNA damaging agents by modulating the PERK pathway [49]. Although further studies are still needed, these results suggest that TRAP1 might have a critical role in the modulation of UPR in the ER. 

Indeed, TRAP1 is emerging as a key regulator of bidirectional crosstalk between ER and mitochondria, having a crucial role in cellular homeostasis maintenance [49]. ER-localized TRAP1 may also contribute to cell survival and resistance to ER stress inducer agents [73] and genotoxic agents [49]. Moreover, ER-associated TRAP1 might be involved in the regulation of the mitochondrial apoptotic pathway by controlling the expression of specific client proteins [49], contributing to cellular resistance to stress conditions.

TRAP1 also has several interactions taking place in the cytoplasm [21]. Chen et al. reported, using immunostaining and cellular fractionation in CV1 kidney cells, that TRAP1 localizes in the cytoplasm and distributes to a specific cytoplasmic region immediately surrounding the nucleus [74]. However, these subcellular fractionation experiments should be optimized, for example, by studying highly selective markers in TRAP1-containing fractions. These refinement of procedures and accurate analysis are crucial to better discriminate TRAP1 localization and role outside the mitochondria. The authors also showed that TRAP1 interacts with Retinoblastoma protein (Rb) during mitosis and after heat shock. Additionally, they found that TRAP1 refolds denatured Rb to its native conformation in vitro. This is of special importance considering that Rb plays a vital role in cell cycle progression and cellular differentiation [74]. 

Song et al. reported that TRAP1 binds to the intracellular domain of TNFR1 [17]. However, TRAP1 is probably a misnomer, as its association with TNFR1 was observed with a two-yeast hybrid assay, which is prone to false-positive results, and it was not confirmed by subsequent studies. Controversies apart, TRAP1 is suggested to interact with the TNF-α/TNFR1 pathway [75,76]. In this pathway, following its interaction with TRAP1, TNFR1 phosphorylates signal transducer and activator of transcription 3 (STAT3) that initiates transcription of the E2F1 transcription factor [68]. Kubota et al. demonstrated, in a neuronal cell line, that, under *TRAP1* knockdown conditions, there is a decrease in phosphorylation of STAT3, which results in the reduction of E2F1 and a downregulation of N-cadherin. This affects the adhesive properties of the cells and influences the morphology of dendritic spines. 

## 4. TRAP1 and Neurodegeneration

Parkinson’s disease is a neurodegenerative disorder attributed to the loss of dopaminergic neurons in the substantia nigra in the midbrain. Mitochondrial dysfunction is observed in autosomal recessively inherited forms of Parkinson’s disease caused by mutations in PINK1, PARKIN, DJ-1, and (HtrA Serine Peptidase 2) HTRA2 genes [77,78,79,80]. Fitzgerald et al. reported a homozygous p.Arg47Ter single nucleotide exchange (R47X) in exon 2 of *TRAP1*, that leads to a premature stop codon and truncation at the transit sequence of TRAP1, in a late-onset Parkinson’s disease patient [81]. It is still under debate if *TRAP1* is a causative gene of Parkinson’s disease [82,83], and therefore further research is required. Controversy apart, the authors demonstrated that fibroblasts from the R47X patient present increased mitochondrial turnover and respiration, increased complex I activity and ATP output, and reduced mitochondrial membrane potential [81].

Several *Drosophila* transgenic lines were generated and used to study the role of TRAP1 in neurons [84,85,86,87,88]. Loss of TRAP1 leads to weak mortality (≈10%) in early time points without affecting the flies life span [84,85]. However, *Drosophila Trap1* mutants present motor impairment [85,87], lower brain dopamine and higher serotonin levels compared to wild-type flies [87]. Ablation of *Drosophila Trap1* increases the sensitivity to stress caused by heat, paraquat, rotenone or antimycin [85,87], and the production of mitochondrial ROS in optic lobes of young adult flies’ brains [85] and in-flight muscle from fly thoraces [84]. On the other hand, others showed that mutation or knockdown of *Trap1* markedly enhanced *Drosophila* survival under oxidative stress, caused by paraquat or rotenone [84]. Depletion of TRAP1 induces the mitochondrial unfolded protein response (UPRmt), a protective response pathway between mitochondria and the nucleus that is initiated in response to a mitochondrial stress signal [84,89] (Figure 5). 

TRAP1 plays a role in the PINK1/PARKIN signaling pathways, as it is phosphorylated and activated by PINK1 [64]. Interestingly, *Trap1* overexpression rescue the phenotypical features observed in *Parkin* mutant flies by improving the survival rate, decreasing the degree of thoracic indentations [87] and partially attenuating wing posture defects [86]. However, *Trap1* expression fails to rescue the climbing defect of the *Parkin* mutants [86,87], reduced ATP levels in the thorax, impaired mitochondrial integrity and declined the levels of complex I subunits [86]. The same authors demonstrated that TRAP1 is also unable to rescue mitochondrial fragmentation and dysfunction upon siRNA-induced silencing of *PARKIN* in human neuronal SH-SY5Y cells [86].

Expression of *Trap1* seems to be sufficient to revert the phenotype of *Pink1* mutant flies, including thoracic indentations, motor impairment, decreased lifespan, and mitochondrial dysfunction [87]. Others demonstrated that genetic and pharmacological TRAP1 inhibition, and not overexpression, in *Drosophila* markedly enhances survival rate under oxidative stress and rescues mitochondrial dysfunction and dopaminergic (DA) neuronal loss induced by *Pink1* mutation [84]. Specific expression of *Trap1* in neurons seems sufficient to suppress neurodegeneration and muscle degeneration and reverse the respiration deficit present in *Pink1* null or mutant flies [21,24]. 

Recently, TRAP1 was described to interact with HTRA2, and its overexpression rescued mitochondrial phenotypes associated with a loss of HTRA2 function, suggesting a signaling pathway downstream of PINK1 and HTRA2 [81].

The formation of abnormal aggregates in the brain of an α-synuclein protein, called Lewy bodies, is one of the pathological hallmarks of Parkinson’s disease [90]. Human α-synuclein mutation or overexpression results in cytotoxicity, with [A53T]α-synuclein being the most toxic variant known [88]. Decreased *Trap1* expression in [A53T]α-synuclein–expressing flies further enhances the loss of dopaminergic neuron number, the loss of climbing ability and increases their sensitivity to oxidative stress [88]. Moreover, overexpression of TRAP1 counteracts [A53T]α-synuclein induced mitochondrial stress in flies, rat primary neurons and SH-SY5Y human neuronal cells. The same study suggests that the ATPase domain is required for TRAP1 mediated protection [88].

Several studies demonstrated the impact of the ablation or knockdown of TRAP1 in vitro, mainly using the SH-SY5Y neuroblastoma cell line [68,86,91,92]. *TRAP1* knockdown in SH-SY5Y cells resulted in abnormal mitochondrial morphology, with decreased number of fragmented mitochondria and increased connectivity. TRAP1 silencing led to decreased levels of dynamin-related protein 1 (Drp1) and mitochondrial fission factor (Mff), two mitochondrial fission proteins, via a nontranscriptional mechanism. However, the expression levels of other fission (Fis1 and mitochondrial dynamics of 51 kDa protein (MiD51)/mitochondrial elongation factor 1 (MIEF1)) and fusion (mitofusin 1-2 (Mfn1/2) and protein optic atrophy protein (OPA)) proteins were unchanged, suggesting that TRAP1 plays a role in mitochondrial fusion/fission, which affects mitochondrial/cellular function [91]. Kubota et al. demonstrated that knockdown of *TRAP1* using small interfering RNA decreases tyrosine phosphorylation of STAT3, followed by a reduction of the transcription factor E2F1, resulting in a downregulation of N-cadherin, and consequently reducing the adhesive properties of the cells. In addition, in cultured rat hippocampal neurons, reduced expression of N-cadherin by *TRAP1* knockdown influences the morphology of dendritic spines increasing the number of sessile spines, spines with reduced stalk construction [68]. Treatment of mouse dopaminergic neuronal (MN9D) cells with 6-hydroxydopamine (6-OHDA) promotes TRAP1 release from the mitochondria into the cytosol [93].

Using in vitro and in vivo models of ischemic injury, the Giffard laboratory demonstrated a protective effect mediated by TRAP1 [94,95]. TRAP1 overexpression in primary astrocyte mouse cultures decreases ROS production, preserves mitochondrial membrane potential during glucose deprivation, and preserves ATP levels and cell viability during oxygen-glucose deprivation [94]. In addition, in vivo overexpression of TRAP1 in neurons and astrocytes, by DNA transfection, reduced infarct area and improved neurological outcome significantly. This was associated with improved mitochondrial function, increased preservation of ATP levels in the brain, and reduction of free radical generation and lipid peroxidation [95]. 

The literature supports the important role of TRAP1 in neurodegeneration opening the perspective of therapeutical approaches based on the modulation of this protein. 

## 5. TRAP1 Inhibitors and Neuroprotection

Due to the similarities between HSP90 family members, the development of TRAP1 specific inhibitors was a demanding task [60,96]. The difficulty of the task is exacerbated by the need for the inhibitor to accumulate in sufficient amount in the mitochondria, where most of the TRAP1 resides. Besides the difficulties, several TRAP1 inhibitors were developed and tested as a treatment for different tumors (Table 1). 

The rationale is that inhibition of TRAP1 would decrease the tumor ability to inhibit apoptosis, tumor resistance to oxidative stress and mitochondrial dysfunction. 

To improve the penetration of gamitrinibs in the mitochondria the 17-(allylamino)-17-demethoxygeldanamycin (17-AAG) region was linked to a mitochondrial targeting sequence of either one to four tandem repeats of cyclic guanidinium (gamitrinib-(G1-G4)) or a triphenylphosphonium (gamitrinib-TPP) [62]. Gamitrinibs are non-specific inhibitors of TRAP1 once these also act on mitochondrial HSP90 and mediate cell death via CyD-dependent mPTP opening [62]. A similar approach was used to create SMTIN-P01 by replacing the isopropylamine of the HSP90 inhibitor PU-H71 with the mitochondria-targeting moiety triphenylphosphonium (TPP) [34]. Hu et al. synthesized a series of TPP-conjugated SMTIN-P01 analogues, substituting the C6 linker with carbon chains of different lengths. From these, the compound with a 10-carbon linker (SMTIN-C10) presented stronger inhibition of TRAP1 than SMTIN-P01 [106].

Shepherdin is a cell-penetrating peptide mimetic able of accumulating in mitochondria and inhibit both TRAP1 and HSP90 [97,98]. To achieve prolonged Shepherdin production and release, researchers tested delivery of Shepherdin using adeno-associated virus (AAV) vectors [107,108].

DN401 (compound 4) is a modification of the indole ring of the HSP90 inhibitor BIIB021 to pyrazolo-pyrimidin. This inhibitor is a mitochondria-permeable TRAP1 inhibitor without a mitochondria delivery vehicle, which showed potent antitumor activity in vitro and in vivo. However, it can simultaneously inactivate other HSP90 family proteins [71,100]. Structure-based drug design guided the optimization of the potency of the DN401 molecule, leading to the identification of compounds 47 and 48 as potent TRAP1 and HSP90 inhibitors. X-ray cocrystallization studies confirmed both 47 and 48 interact with the ATP binding pocket in the TRAP1 protein [101]. 

Unlike the above inhibitors, 2-amino-7,8-dihydro-6H-pyrido[4,3-D]pyrimidin-5-one (NVP-HSP990) is an oral TRAP1 inhibitor and also inhibits the activity of HSP90, which is structurally classified as an aminomididine. NVP-HSP990 was tested in phase I clinical trials (NCT01064089 and NCT00879905) for the treatment of advanced solid malignancies. NVP-HSP990 was relatively well tolerated, with neurological toxicity being the most relevant determine dose-limiting toxicities [102,103,104]. 

Honokiol dichloroacetate ester (HDCA) is a vegetal derivative honokiol and its lipophilic bis-dichloroacetate ester, able to bind an allosteric site in TRAP1, which inhibits TRAP1 but not HSP90 ATPase activity. HDCA leads to oxidative stress and apoptosis in vivo tumor models and displays an action that is functionally opposed to that of TRAP1, as it induces both succinate dehydrogenase and the mitochondrial deacetylase sirtuin-3 (SIRT3), which further enhances succinate dehydrogenase activity [105]. 

Recently, Giorgio Colombo’s research team used a dynamics-based approach to identify a TRAP1 allosteric pocket distal to its active site that can host drug-like molecules, this led to the identification of eleven small molecules with the optimal stereo-chemical features (Table 1). These 11 small molecules inhibit TRAP1, but not HSP90 or ATPase activity, and they revert TRAP1-dependent downregulation of succinate dehydrogenase activity in human and mouse malignant peripheral nerve sheath tumor cells and zebrafish [52]. 

The potential of different HSP90 inhibitors such as 17-AAG, 17-DMAG, and TAS116 as a treatment for neurodegenerative disorders and their neurotoxicity were previously evaluated [109,110,111,112,113,114,115,116,117,118]. There are very few studies describing the effect of the TRAP1/HSP90 inhibitors, listed in Table 1, on neuronal cells. Here, we summarized the present literature. Gamitrinib-TPP seems to be toxic to primary mouse neurons even at a low concentration (5 µM). In neuronal cultures obtained by directly converting skin fibroblasts, Gamitrinib-TPP mediated activation of PINK1/PARKIN inducing mitophagy [119]. Chronic oral administration of NV-HSP990 improves huntingtin aggregate load, motor performance and maintaining body and brain weight in the R6/2 mouse model of Huntington’s disease. However, the beneficial effects of this treatment were transient and diminished with disease progression [120]. The same authors showed that systemic NV-HSP990 administration activates the heat shock responses in mouse brain tissue [120]. Both HSP70 and HSP90 are present in the retina [121,122]. In human trials, some HSP90 inhibitors, including 17-DMAG and NVP-AUY922, caused visual disorders indicative of retinal dysfunction, while others such as 17-AAG and ganetespib did not [109], which might suggest a role for this chaperone in the retina. On the other hand, administration of NV-HSP990, a TRAP1/HSP90 inhibitor, promotes the preservation of photoreceptor function measured by electroretinography and photoreceptor survival in a rat model of autosomal dominant retinitis pigmentosa, the P23H-1 mutant. Furthermore, the treatment-induced heat shock response and reduced protein aggregation in the P23H-1 rat retina [123]. To the best of our knowledge, TRAP1 expression in the retina was not yet demonstrated, remaining inconclusive if TRAP1 inhibition in this tissue contributes to the protection mediated by NV-HSP990.

## 6. Concluding Remarks and Future Perspectives

The role of TRAP1 was extensively studied in the oncology field due to its abnormal expression in tumors, capacity to promote proliferation and motility, and facilitate the invasion and metastasis of tumor cells. However, in the neurodegeneration field only recently TRAP1 started to attract attention. Recent studies suggest that TRAP1 plays a protective role in Parkinson’s disease and might even be a disease causative gene. The important role of TRAP1 in neurons is further reinforced by the studies using TRAP1/HSP90 inhibitors that induce neuronal toxicity and cause visual disorders. All these new findings suggest a role for TRAP1 in neurons and neurodegenerative processes open new therapeutical approaches for such devastating disorders.

## Figures and Tables

**Figure 1 antioxidants-10-01829-f001:**
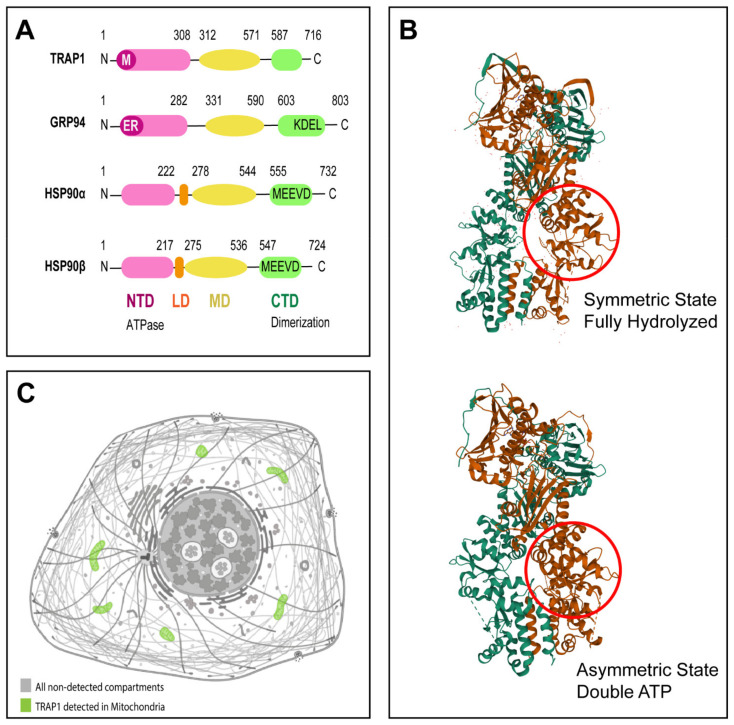
TRAP1 molecular structure. (**A**) Prototypic HSP90 family member has three structural domains, an N-terminal domain (NTD) responsible for ATP binding, a middle domain (MD) which is important for client binding, and a C-terminal domain (CTD) responsible for dimerization. GRP94 contains also a KDEL motif. While HSP90α and HSP90β contain a MEEVD motif and a linker domain (LD). Mitochondrial signal sequence present in TRAP1 is indicated with “M” and endoplasmic reticulum signal sequence present in GRP94 is indicated with “ER”. Lengths of HSP90α, HSP90β, GRP94, and TRAP1 are 732, 724, 803, and 704 amino acids, respectively. (**B**) 3D structures of TRAP1 dimer in different nucleotide states, with each protomer in a different color. Symmetric state of TRAP1 in fully hydrolyzed state of protein (PDB code 5TVX; image available at https://www.rcsb.org/3d-view/5TVX/1; accessed on 29 September 2021) [11,12,13,14]. Asymmetric state of TRAP1 in double ATP state (PDB code 6XG6; image available at https://www.rcsb.org/3d-view/6XG6/1; accessed on 29 September 2021) [11,13,14,15]. Red circle highlights substructure in middle domain that undergoes rearrangement upon nucleotide hydrolysis; reproduced with permission from RCSB PDB. (**C**) TRAP1 is primarily located in mitochondria. TRAP1 detection in cell mitochondria is highlighted in green (image available at v20.proteinatlas.org/ENSG00000126602-TRAP1/cell; accessed on 29 September 2021); reproduced with permission from The Human Protein Atlas.

**Figure 2 antioxidants-10-01829-f002:**
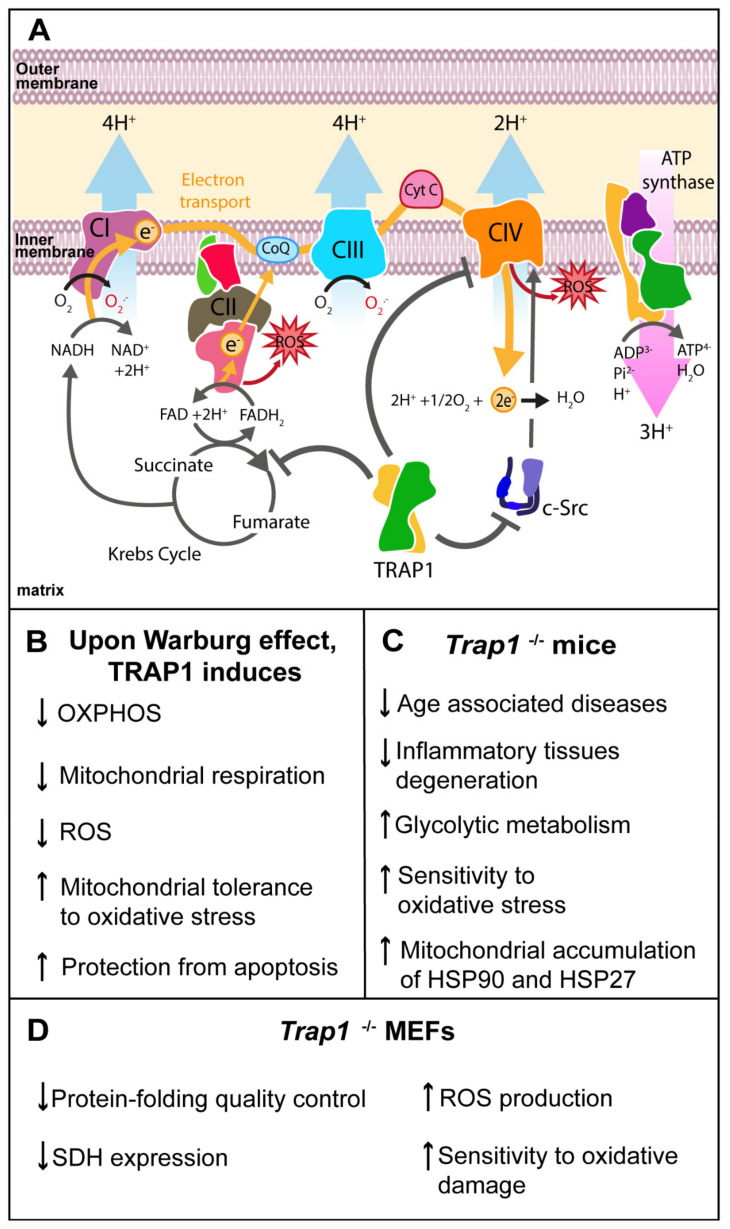
Role of TRAP1 in energetic metabolism regulation. (**A**) TRAP1 modulates oxidative phosphorylation (OXPHOS) by interacting with proteins of mitochondrial electron transport chain (ETC), namely succinate dehydrogenase (SDH) and cytochrome oxidase, also known as complex II and complex IV, respectively. TRAP1 inhibits proto-oncogene tyrosine-protein kinase (c-Src) downregulating cytochrome oxidase activity. TRAP1 also inhibits SDH activity, resulting in less electron funneling and lower reactive oxygen species (ROS) generation as a side-reaction of ETC. (**B**) Upon conditions of shortage of oxygen, also known as Warburg effect, TRAP1 induces downregulation of OXPHOS, decreasing ROS production and mitochondrial respiration. Moreover, OXPHOS downregulation is associated with increased mitochondrial tolerance to oxidative stress, and protection from apoptosis; (**C**) *Trap1* knockout mouse showed a reduced incidence of age-associated pathologies and inflammatory tissue degeneration. It also presented a switch to glycolytic metabolism, increased susceptibility to oxidative stress and increased mitochondrial accumulation of cytoprotective chaperones Hsp90 and Hsp27. (**D**) *Trap1* knockout mouse embryonic fibroblasts (MEFs) exhibited loss of protein-folding quality control in mitochondria and decreased SDH expression. While showing increased ROS production and increased sensitivity to oxidative stress.

**Figure 3 antioxidants-10-01829-f003:**
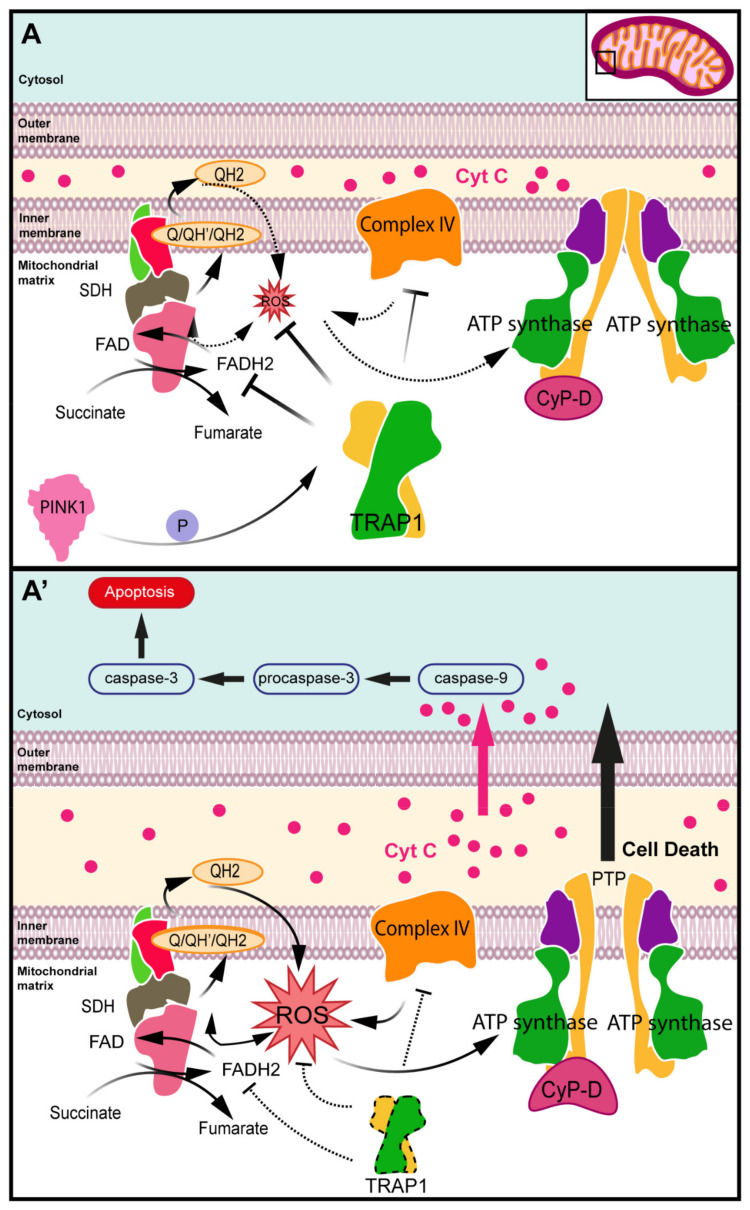
Role of TRAP1 in redox homeostasis. (**A**) TRAP1 inhibits reactive oxygen species (ROS) accumulation by controlling respiratory complexes of electron transport chain (ETC), namely, succinate dehydrogenase (SDH) and cytochrome oxidase. Through its interaction with cyclophilin D (CypD), TRAP1 inhibits mitochondrial permeability transition pore (mPTP) formation. Overexpression of TRAP1 blocks mitochondria-mediated apoptotic cascade and ensuing caspase-3 activation. (**A’**) PTEN-induced kinase 1 (PINK1) depletion leads to cytochrome C (CytC) release, which correlates with reduction in TRAP1 phosphorylation by PINK1. Activated CypD sensitizes mPTP to opening, leading to mitochondrial swelling and depolarization, with outer membrane ruptures causing release of CytC into cytosol and eventually cell death.

**Figure 4 antioxidants-10-01829-f004:**
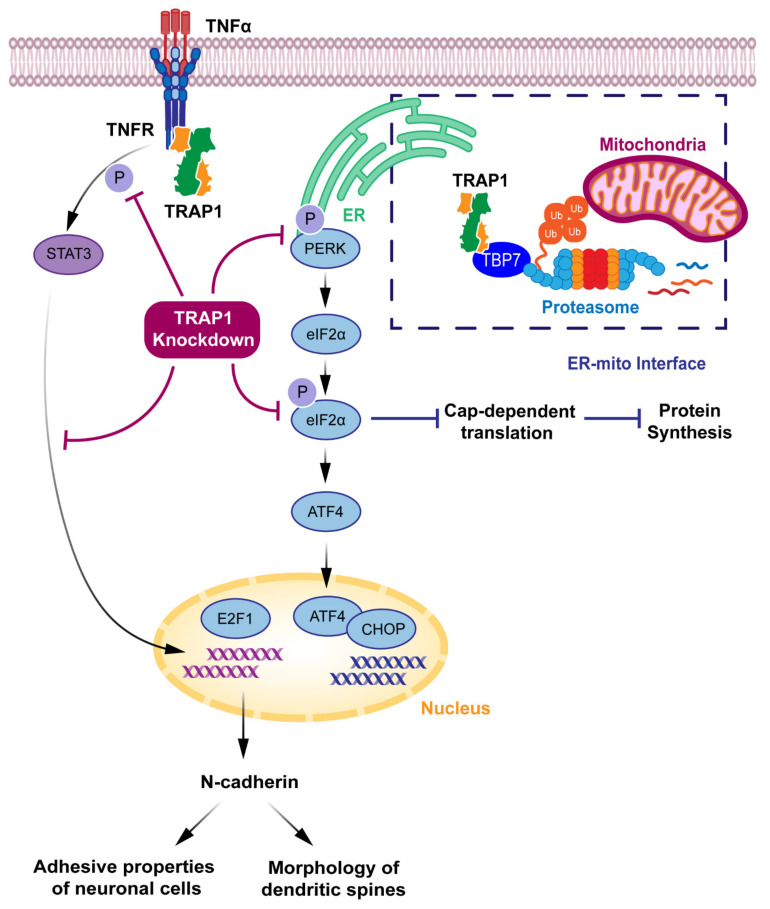
TRAP1 suggested extra-mitochondrial roles and proposed mechanisms. In endoplasmic reticulum (ER)-mitochondria interface, TRAP1 is associated with tat-binding protein 7 (TBP7), controlling ubiquitination and mitochondrial protein folding. In cytoplasm, TRAP1 binds to intracellular domain of tumor necrosis factor receptor-1 (TNFR1), thus interacting with tumor necrosis factor-α (TNF-α)/TNFR1 pathway. TNFR1 phosphorylates signal transducer and activator of transcription 3 (STAT3) that activates transcription factor E2F1. *TRAP1* knockdown decreases phosphorylation of STAT3, which results in reduction of E2F1 and a downregulation of N-cadherin, responsible for adhesive and morphological properties of neuronal cells. Activation of selective stress-responsive protein RNA-like ER kinase (PERK), and phosphorylation of translation factor eukaryotic initiation factor-2α (eIF2α) block cap-dependent translation, thus reducing global protein synthesis in cell. *TRAP1* knockdown decreases PERK and eIF2α phosphorylation, and expression of C/EBP homologous protein (CHOP).

**Figure 5 antioxidants-10-01829-f005:**
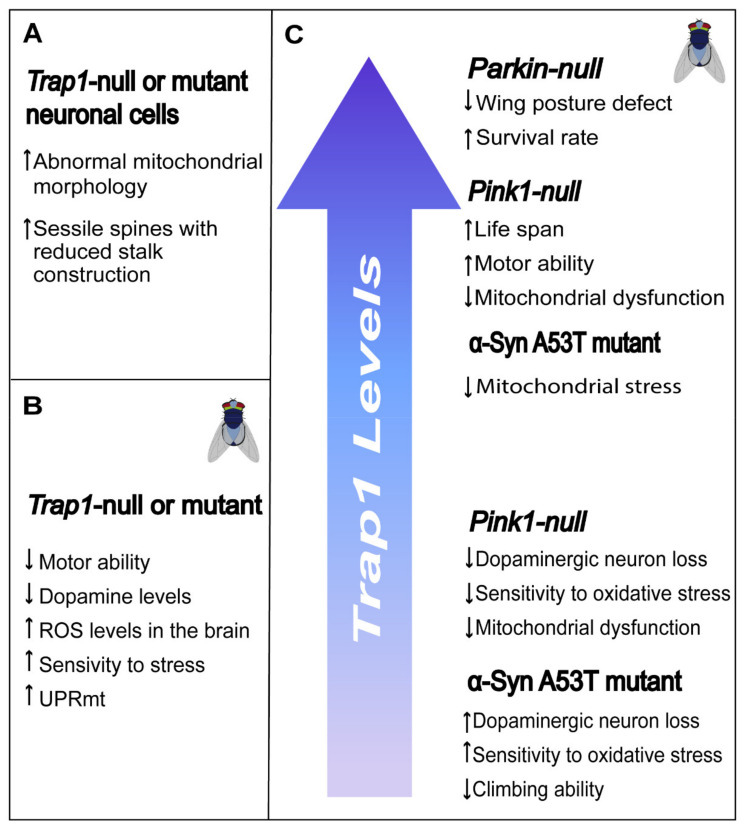
Role of TRAP1 in neurodegeneration. (**A**) SH-SY5Yneuroblastoma cells lacking TRAP1 showed abnormal mitochondrial morphology, with decreased number of fragmented mitochondria with increased connectivity. Moreover, hippocampal rat neurons with reduced levels of TRAP1 presented an increased number of sessile spines with reduced stalk construction. (**B**) *Trap1*-null or mutant flies presented decreased motor ability and dopamine levels, while showing increased reactive oxygen species (ROS) levels in their brains, increased sensitivity to stress, and increased mitochondrial unfolded protein response (UPRmt). (**C**) Knockdown or overexpression of *TRAP1* in different genetic backgrounds has different outcomes in terms of survival of dopaminergic neurons, mitochondrial dysfunction, life span, and motor ability.

**Table 1 antioxidants-10-01829-t001:** List of TRAP1 inhibitors.

Drug	Properties	References
Gamitrinib-(G1-G4) and Gamitrinib-TPP	TRAP1 and HSP90 inhibitors	[62]
Shepherdin	TRAP1 and HSP90 inhibitor	[97,98]
SMTIN-P01	TRAP1 inhibitor	[34]
SMTIN-C10	TRAP1 inhibitor	[99]
DN401 (compound 4)	Hsp90 paralogs, including TRAP1	[71,100]
Compounds 47 and 48	TRAP1 and HSP90 inhibitors	[101]
NVP-HSP990	TRAP1 and HSP90 inhibitor	[102,103,104]
HDCA	TRAP1 inhibitor	[105]
Compound 1 Source: Vitas-M; Cat#STK031415	TRAP1 inhibitors	[52]
Compound 2 Source: Enamine; Cat#Z1128779798		
Compound 3 Source: National Cancer Institute (NCI); Cat#NSC338501; CAS: 26988-58-9		
Compound 4 Source: NCI; Cat#NSC668594		
Compound 5 Source: Ambinter; Cat#AMB9798487		
Compound 6 Source: Enamine; Cat#Z363507628		
Compound 7 Source: NCI; Cat#NSC56914; CAS: 6947-27-9		
Compound 8 Source: Ambinter; Cat#AMB3429185		
Compound 9 Souce: Vitas-M; Cat#STL380969		
Compound 10 Source: NCI; Cat#NSC1032; CAS: 5336-09-4		
Compound 11 Source: NCI; Cat#NSC151831		

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
