# Peer review of "TRAP1 in Oxidative Stress and Neurodegeneration"

_antioxidants, 2021, doi:10.3390/antiox10111829_

Round 1

Reviewer 1 Report

The MS: antioxidants-1423475 Rego et al is dealing with the physiological and pathological role of TRAP1 (HSP75) protein in oxidative stress and neurodegeneration.  The review’s structure is clear and sound. Authors discuss the structure, localization (mitochondrial and nonmitochondrial) of TRAP protein. The comprehensive description of TRAP1-associated bioenergetical  changes is excellent. The role of TRAP1 in tumor associated reprogramming of metabolism is particularly interesting. The review gives a detailed description of TRAP1’s role in neurodegeneration mainly focusing on Parkinson’s disease. The schematic Figures are excellent, references are correctly cited.

The referee has only some minor comments questions and suggestions.

row 69-70 “the affinity of TRAP1 for ATP is  one order of magnitude higher” It would be interesting to give concentrations as well.

row 96-98 „The relative preference of TRAP1 for Ca2+ or Mg2+ is dependent upon the free ATP concentration, with higher ATPase activity with Ca2+ at high ATP concentrations [18]  (Figure 1).” Again giving ATP concentrations could be very important for “mitochondriacs” like the referee.

Fig.5. The legend does not reflect the figure.  A and B are reversed.

The numbering of the subchapters of the MS are confusing. there are two “4” subchapters:

“TRAP1 and neurodegeneration” and “Concluding remarks”

Author Response

  1. row 69-70 “the affinity of TRAP1 for ATP is one order of magnitude higher” It would be interesting to give concentrations as well.

Answer: As requested by reviewer 1, the following text was added: “Indeed, the Michaelis constant (Km) of TRAP1 for ATP binding is 14.3 µM, much lower than the average Km of 127 µM for the other HSP90 paralogs [22].”

  1. row 96-98 „The relative preference of TRAP1 for Ca2+ or Mg2+ is dependent upon the free ATP concentration, with higher ATPase activity with Ca2+ at high ATP concentrations [18]  (Figure 1).” Again giving ATP concentrations could be very important for “mitochondriacs” like the referee.

Answer: We added the requested information to the text: “The relative preference of TRAP1 for Ca2+ or Mg2+ is dependent upon the free ATP concentration, with higher ATPase activity with Ca2+ at high ATP concentrations (350 µM) and higher ATPase activity with Mg2+ for low ATP concentrations (35 µM) [18] (Figure 1).”

  1. 5. The legend does not reflect the figure.  A and B are reversed.

Answer: We have corrected the figure legend.

  1. The numbering of the subchapters of the MS are confusing. there are two “4” subchapters: “TRAP1 and neurodegeneration” and “Concluding remarks”.

Answer: We apologize for the error. We replace the number 4 with a 6 on concluding remarks.

Reviewer 2 Report

In the present manuscript, Rego et al analyze the functions of the mitochondrial chaperone TRAP1, with a particular focus on its activity has an anti-oxidant and anti-apoptotic molecule, as well as on its role in neurodegenerative disorders. In my opinion, several points must be better addressed and explained in order to improve the quality of the text:

line 60: TRAP1 expression in physiological conditions is not limited to brain and testis (see https://www.proteinatlas.org/ENSG00000126602-TRAP1/tissue)

line 67: it is incorrect to state that “inhibition of the N-terminal ATPase of TRAP1 supports tetramer formation”, as the interplay between tetramerization and activity remains largely unexplored

line 72: the sentence “the ATPase activity is inversely correlated with client recognition by TRAP1” is unclear to me.

line 129: English is unclear. The Authors probably mean “inhibition of SDH activity IS enhanced by TRAP1 ERK1/2-dependent phosphorylation AND has antioxidant and antiapoptotic effects in tumour cells”.

Figure 2: OXPHOS depiction is incomplete. Protons, oxygen and water must be added, and even more importantly, succinate, as it is the driver of the TRAP1-dependent HIF1alpha stabilization discussed in the text. In B, “TRAP1 favours” is ambiguous and could be corrected with “TRAP1 induces”. In C, pathology is the study of a disease, not the disease itself.

line 158: analysis of the context-dependent roles of TRAP1 on cell bioenergetics is unclear, as all reported observations in the text point to its inhibition of OXPHOS. Probably the Authors refer to controversies aboutTRAP1 role in tumorigenicity, but this must be clearly explained.

line 162: energy is never “produced” according to thermodynamic laws.

line 195: exploited?

line 197: cyt c release cannot be “eccessive”, as it is released only during cell death induction

line 208 and following: the interplay between Ca2+, PTP opening, CyPD and TRAP1 is explained in a confused way. Please rewrite it remaining strictly connected to the observations reported in literature. The same applies to Fig. 3 (e.g.: cyt c and caspases are not in the mitochondrial matrix, but in the intermembrane space and in the cytosol, respectively). Importantly, no connection between PINK1-dependent phosphorylation of TRAP1 and TRAP1-dependent interaction with CyPD has ever been reported (Figure 3 and line 234). Similarly, no interaction between TRAP1 and cyt c is known (line 233). TRAP1 cannot inhibit PTP opening by decreasing caspase activity (line 240), as caspase activation is downstream to PTP opening

Chapter 3.2: there is a wide controversy on the possibility that TRAP1 could act outside mitochondria, as stated by the same Authors in Fig. 1C. In my opinion, the Authors should highlight this, understating the discussion on TRAP1 possible functions outside mitochondria, as further data are probably needed to surely determine this point. For instance, it is important to underline that accurate analyses (i.e., study of highly selective markers in TRAP1-containing fractions) are required to separate ER and mitochondria components in subcellular fractionation experiments. When it is affirmed that TRAP1 is localized in the ER, these analyses must be performed, and this is not always the case in the papers reported in the text. This must be checked by the Authors, in order to better discriminate sound data. Importantly, TRAP1 is probably a misnomer, as its association with TNF-receptor (line 355) was observed with a two-yeast hybrid assay, which is prone to false positive results, and never confirmed in subsequent studies. Moreover, several extra-mitochondrial effects of TRAP1 are merely correlative (e.g. changes in eIF2alpha phosphorylation, or in ER stress proteins, in lines 315-340), and this must be underlined by the Authors. The Authors must be careful all along the text to discriminate mechanistic connections in which TRAP1 is involved from simple correlations, using the appropriate verbs to avoid overinterpretations (e.g. “TRAP1 determines” is correct only when its causative effect is demonstrated).

line 438: the suggestion that “TRAP1 controls mitochondrial fusion/fission” is an overstatement.

line 470: this sentence must be moved downward, after gamitrinib structure is described.

In Chapter 5, a mention to recently developed TRAP1 allosteric inhibitors (doi: 10.1016/j.celrep.2020.107531) is missing.

line 533: “studied” instead of “study”

Author Response

  1. line 60: TRAP1 expression in physiological conditions is not limited to brain and testis (see https://www.proteinatlas.org/ENSG00000126602-TRAP1/tissue).

Answer: We updated our text accordingly to the data present on the suggested website: “TRAP1 is expressed in several tissues, including the central nervous system (https://www.proteinatlas.org/ENSG00000126602-TRAP1/tissue) [12–14].”

  1. line 67: it is incorrect to state that “inhibition of the N-terminal ATPase of TRAP1 supports tetramer formation”, as the interplay between tetramerization and activity remains largely unexplored

Answer: Accordingly, to the comment of reviewer 2, we have removed this statement.

  1. line 72: the sentence “the ATPase activity is inversely correlated with client recognition by TRAP1” is unclear to me.

Answer: Taking into consideration the comment of reviewer 2, we have removed this statement.

  1. line 129: English is unclear. The Authors probably mean “inhibition of SDH activity IS enhanced by TRAP1 ERK1/2-dependent phosphorylation AND has antioxidant and antiapoptotic effects in tumour cells”.

Answer: We corrected the sentence accordingly to the reviewer 2 suggestion: “The inhibition of SDH activity is enhanced by TRAP1 ERK1/2-dependent phosphorylation and has antioxidant and antiapoptotic effects in tumour cells, stabilizing the hypoxia-inducible factor 1α (HIF1α), a transcription factor required for tumour cell growth [26,36,39]”

  1. Figure 2: OXPHOS depiction is incomplete. Protons, oxygen and water must be added, and even more importantly, succinate, as it is the driver of the TRAP1-dependent HIF1alpha stabilization discussed in the text. In B, “TRAP1 favours” is ambiguous and could be corrected with “TRAP1 induces”. In C, pathology is the study of a disease, not the disease itself.

Answer: We added the requested elements to figure 2, including protons, oxygen, water and succinate. Moreover, we have also corrected the text in the figure as suggested by this review.

  1. line 158: analysis of the context-dependent roles of TRAP1 on cell bioenergetics is unclear, as all reported observations in the text point to its inhibition of OXPHOS. Probably the Authors refer to controversies aboutTRAP1 role in tumorigenicity, but this must be clearly explained.

Answer: Taking into consideration the comment of reviewer 2 and the fact that our text is focused on OXPHOS inhibition and not tumorigenicity, we decided to remove this statement.

  1. line 162: energy is never “produced” according to thermodynamic laws.

Answer: Taking into consideration the comment of reviewer 2, we adapted the sentence: “OXPHOS sustains organelle function and plays a central role in cellular energy metabolism, with increased energy efficiency when compared to glycolysis.”

  1. line 195: exploited?

Answer: The word “exploited” was removed.

  1. line 197: cyt c release cannot be “eccessive”, as it is released only during cell death induction

Answer: We deleted the word “excessive”.

  1. line 208 and following: the interplay between Ca2+, PTP opening, CyPD and TRAP1 is explained in a confused way. Please rewrite it remaining strictly connected to the observations reported in literature. The same applies to Fig. 3 (e.g.: cyt c and caspases are not in the mitochondrial matrix, but in the intermembrane space and in the cytosol, respectively). Importantly, no connection between PINK1-dependent phosphorylation of TRAP1 and TRAP1-dependent interaction with CyPD has ever been reported (Figure 3 and line 234). Similarly, no interaction between TRAP1 and cyt c is known (line 233). TRAP1 cannot inhibit PTP opening by decreasing caspase activity (line 240), as caspase activation is downstream to PTP opening

Answer: We are grateful for the pertinent comment that helps us to improve this section. Taking into consideration the comment of reviewer 2, we adapted our text, remaining strictly connected to what is reported in the literature and clarifying some ideas and relations: “Mitochondrial matrix Ca2+ content is known to be a key factor for the regulation of the transitions between an open and closed state of the mitochondrial permeability transition pore (mPTP) [58–60]. However, there is no obvious correlation between the concentration of mitochondrial matrix Ca2+ and the onset of mPTP opening. In fact, it is thought that the triggering of mPTP opening is not caused by Ca2+ overload itself, but by additional factors that still need to be characterized [59]. Since mPTP opening results in mitochondrial inner membrane potential loss, membrane rupture and apoptosis [2,10,57], mPTP regulation is critical for mitochondrial homeostasis [56]. The mitochondrial matrix protein cyclophilin D (CypD) is known to be a regulator of the mPTP [44,49,59]. Indeed, CypD-deficient mitochondria generate fewer ROS and are less susceptible to Ca2+-induced mitochondrial swelling and permeability transition.”;

We have also made some changes/corrections in Figure 3 and the respective legend: “The role of TRAP1 in redox homeostasis. (A) TRAP1 inhibits ROS accumulation by controlling respiratory complexes of the electron transport chain (ETC), namely succinate dehydrogenase (SDH) and cytochrome oxidase. Through its ATPase activity, TRAP1 inhibits conformational changes of CypD, thus blocking mitochondrial permeability transition pore (mPTP) components conformational switch and consequent pore formation. Overexpression of TRAP1 prevents apoptosis by decreasing caspase-3 activity, thus blocking the mitochondria-mediated apoptotic cascade. (A’) PINK1 depletion leads to CytC release, which correlates with the reduction in TRAP1 phosphorylation by PINK1.  Activated CypD is responsible for the conformational switch of mPTP components, creating non-selective pores in the mitochondrial inner membrane. This leads to mitochondrial swelling, transmembrane potential loss, mitochondrial rupture and apoptosis. CypD is also responsible for the release of cytochrome C (CytC) into the cytosol, thus inducing cell death. Release of CytC can also result from excessive accumulation of ROS resultant from ETC side-reactions.”

  1. Chapter 3.2: there is a wide controversy on the possibility that TRAP1 could act outside mitochondria, as stated by the same Authors in Fig. 1C. In my opinion, the Authors should highlight this, understating the discussion on TRAP1 possible functions outside mitochondria, as further data are probably needed to surely determine this point. For instance, it is important to underline that accurate analyses (i.e., study of highly selective markers in TRAP1-containing fractions) are required to separate ER and mitochondria components in subcellular fractionation experiments. When it is affirmed that TRAP1 is localized in the ER, these analyses must be performed, and this is not always the case in the papers reported in the text. This must be checked by the Authors, in order to better discriminate sound data. Importantly, TRAP1 is probably a misnomer, as its association with TNF-receptor (line 355) was observed with a two-yeast hybrid assay, which is prone to false positive results, and never confirmed in subsequent studies. Moreover, several extra-mitochondrial effects of TRAP1 are merely correlative (e.g. changes in eIF2alpha phosphorylation, or in ER stress proteins, in lines 315-340), and this must be underlined by the Authors. The Authors must be careful all along the text to discriminate mechanistic connections in which TRAP1 is involved from simple correlations, using the appropriate verbs to avoid overinterpretations (e.g. “TRAP1 determines” is correct only when its causative effect is demonstrated).

Answer: Again, we thank the reviewer for the useful comment, that allowed a clear improvement in this section. Taking into consideration the comment of reviewer 2, we made several adjustments to our text. We have highlighted the controversy in TRAP1 proposed extra-mitochondrial roles, discussing the need for further studies. We have also checked the observations reported in the cited papers to avoid ambiguity or overinterpretations and to discriminate connections from correlations. Please see all the changes in the version with track records.

  1. line 438: the suggestion that “TRAP1 controls mitochondrial fusion/fission” is an overstatement.

Answer: We have adapted the sentence:suggesting that TRAP1 plays a role in mitochondrial fusion/fission, which affects mitochondrial/cellular function

  1. line 470: this sentence must be moved downward, after gamitrinib structure is described.

Answer: We moved the sentence downward as requested by the reviewer.

  1. In Chapter 5, a mention to recently developed TRAP1 allosteric inhibitors (doi: 10.1016/j.celrep.2020.107531) is missing.

Answer: We added a new paragraph to the text: “Recently, Giorgio Colombo´s research team used a dynamics-based approach to identify a TRAP1 allosteric pocket distal to its active site that can host drug-like molecules, this led to the identification of eleven small molecules with the optimal stereo-chemical features (Table 1). These eleven small molecules inhibit TRAP1, but not HSP90, ATPase activity and revert TRAP1-dependent downregulation of succinate dehydrogenase activity in human and mouse malignant peripheral nerve sheath tumor cells and zebrafish [108].” We also added the 11 small molecules indicated in the manuscript to Table 1.

  1. line 533: “studied” instead of “study”

Answer: We replaced “study” with “studied”.

Round 2

Reviewer 2 Report

In the revised version of their manuscript, Rego et al have improved the quality and clarity of the text, which is now an interesting review focused on the roles of the chaperone TRAP1 in redox homeostasis and in cell death regulation, mainly in neurodegenerative diseases.

However, some sentences need to be written in a clearer way. Here are my suggestions:

line 209: “there is no obvious correlation between the concentration of mitochondrial matrix Ca2+ and the onset of mPTP opening. In fact, it is thought that the triggering of mPTP opening is not caused by Ca2+ overload itself, but by additional factors that still need to be characterized[59].”. This should sound more like: “mitochondrial matrix Ca2+overload induces mPTP opening, even though additional factors that are only partially characterized can contribute to pore induction[59].”

Legend to Figure 3: “Through its ATPase activity, TRAP1 inhibits conformational changes of CypD, thus blocking mitochondrial permeability transition pore (mPTP) components conformational switch and consequent pore formation”. Here I suggest this change: “Through its interaction with CypD, TRAP1 inhibits mitochondrial permeability transition pore (mPTP) formation”.

Legend to Figure 3: “Overexpression of TRAP1 prevents apoptosis by decreasing caspase-3 activity, thus blocking the mitochondria-mediated apoptotic cascade”. Here I suggest this change: “Overexpression of TRAP1 blocks the mitochondria-mediated apoptotic cascade and the ensuing caspase-3 activation”.

Legend to Figure 3: “Activated CypD is responsible for the conformational switch of mPTP components, creating non-selective pores in the mitochondrial inner membrane. This leads to mitochondrial swelling, transmembrane potential loss, mitochondrial rupture and apoptosis. CypD is also responsible for the release of cytochrome C (CytC) into the cytosol, thus inducing cell death. Release of CytC can also result from excessive accumulation of ROS resultant from ETC side-reactions.” Here I suggest this change: “Activated CyPD sensitizes PTP to opening, leading to mitochondrial swelling and depolarization, with outer membrane ruptures causing the release of cytochrome C (CytC) into the cytosol and eventually cell death.”

Author Response

We are grateful for the reviewer comments. We have included all the suggested changes in the manuscript. Bellow, we indicated all changes by us performed:

1. line 209: “there is no obvious correlation between the concentration of mitochondrial matrix Ca2+ and the onset of mPTP opening. In fact, it is thought that the triggering of mPTP opening is not caused by Ca2+ overload itself, but by additional factors that still need to be characterized[59].”. This should sound more like: “mitochondrial matrix Ca2+overload induces mPTP opening, even though additional factors that are only partially characterized can contribute to pore induction [59].”

 Answer: We corrected the sentence accordingly to the suggestion. Please see lines 207-209: "Mitochondrial matrix Ca2+ overload induces mPTP opening, even though additional factors that are only partially characterized can contribute to pore induction [59]."

2. Legend to Figure 3: “Through its ATPase activity, TRAP1 inhibits conformational changes of CypD, thus blocking mitochondrial permeability transition pore (mPTP) components conformational switch and consequent pore formation”. Here I suggest this change: “Through its interaction with CypD, TRAP1 inhibits mitochondrial permeability transition pore (mPTP) formation”.

 Legend to Figure 3: “Overexpression of TRAP1 prevents apoptosis by decreasing caspase-3 activity, thus blocking the mitochondria-mediated apoptotic cascade”. Here I suggest this change: “Overexpression of TRAP1 blocks the mitochondria-mediated apoptotic cascade and the ensuing caspase-3 activation”.

 Legend to Figure 3: “Activated CypD is responsible for the conformational switch of mPTP components, creating non-selective pores in the mitochondrial inner membrane. This leads to mitochondrial swelling, transmembrane potential loss, mitochondrial rupture and apoptosis. CypD is also responsible for the release of cytochrome C (CytC) into the cytosol, thus inducing cell death. Release of CytC can also result from excessive accumulation of ROS resultant from ETC side-reactions.” Here I suggest this change: “Activated CyPD sensitizes PTP to opening, leading to mitochondrial swelling and depolarization, with outer membrane ruptures causing the release of cytochrome C (CytC) into the cytosol and eventually cell death.”

Answer: We corrected the legend to Figure 3 accordingly to the suggestions. Please see lines 229-235: "Through its interaction with cyclophilin D (CypD), TRAP1 inhibits mitochondrial permeability transition pore (mPTP) formation. Overexpression of TRAP1 blocks the mitochondria-mediated apoptotic cascade and the ensuing caspase-3 activation. (A’) PINK1 depletion leads to cytochrome C (CytC) release, which correlates with the reduction in TRAP1 phosphorylation by PINK1. Activated CypD sensitizes mPTP to opening, leading to mitochondrial swelling and depolarization, with outer membrane ruptures causing the release of CytC into the cytosol and eventually cell death."